# Tuning Magnetic Properties of a Carbon Nanotube-Lanthanide Hybrid Molecular Complex through Controlled Functionalization

**DOI:** 10.3390/molecules26030563

**Published:** 2021-01-22

**Authors:** Ibwanga S. Mosse, Venkateswara Rao Sodisetti, Christopher Coleman, Siphephile Ncube, Alvaro S. de Sousa, Rudolph M. Erasmus, Emmanuel Flahaut, Thomas Blon, Benjamin Lassagne, Tomas Šamořil, Somnath Bhattacharyya

**Affiliations:** 1Nano-Scale Transport Physics Laboratory, School of Physics, University of the Witwatersrand, Johannesburg Wits 2050, South Africa; sav.mosse@gmail.com (I.S.M.); 2288546@students.wits.ac.za (V.R.S.); Christopher.coleman@wits.ac.za (C.C.); Siphephilen@gmail.com (S.N.); alvarodesousa@gmail.com (A.S.d.S.); 2DSI-NRF Centre of Excellence in Strong Materials and School of Physics, University of the Witwatersrand, Johannesburg Wits 2050, South Africa; Rudolph.Erasmus@wits.ac.za; 3CIRIMAT, Université de Toulouse, CNRS, INPT, UPS, UMR CNRS-UPS-INP No. 5085, Université Toulouse Paul Sabatier, Bât. CIRIMAT, 118, Route de Narbonne, CEDEX 9, 31062 Toulouse, France; emmanuel.flahaut@univ-tlse3.fr; 4Laboratoire de Physique et Chimie des Nano-Objets, Université de Toulouse, UMR 5215 INSA, CNRS, UPS, 135 Avenue de Rangueil, CEDEX 4, F-31077 Toulouse, France; thomas.blon@insa-toulouse.fr (T.B.); benjamin.lassagne@insa-toulouse.fr (B.L.); 5TESCAN Orsay Holding, a.s., Libušina tř. 21, 623 00 Brno, Czech Republic; tomas.samoril@tescan.com

**Keywords:** multi-walled carbon nanotubes, Raman spectroscopy, spintronics

## Abstract

Molecular magnets attached to carbon nanotubes (CNT) are being studied as potential candidates for developing spintronic and quantum technologies. However, the functionalization routes used to develop these hybrid systems can drastically affect their respective physiochemical properties. Due to the complexity of this systems, little work has been directed at establishing the correlation between the degree of functionalization and the magnetic character. Here, we demonstrate the chemical functionalization degree associated with molecular magnet loading can be utilized for controlled tuning the magnetic properties of a CNT-lanthanide hybrid complex. CNT functionalization degree was evaluated by interpreting minor Raman phonon modes in relation to the controlled reaction conditions. These findings were exploited in attaching a rare-earth-based molecular magnet (Gd-DTPA) to the CNTs. Inductively coupled plasma mass spectrometry, time-of-flight secondary ion mass spectrometry and super conducting quantum interference device (SQUID) measurements were used to elucidate the variation of magnetic character across the samples. This controlled Gd-DTPA loading on the CNT surface has led to a significant change in the nanotube intrinsic diamagnetism, showing antiferromagnetic coupling with increase in the Weiss temperature with respect to increased loading. This indicates that synthesis of a highly correlated spin system for developing novel spintronic technologies can be realized through a carbon-based hybrid material.

## 1. Introduction

Since their discovery in 1991 [1], carbon nanotubes (CNTs) have remained one of the most widely researched materials due to their outstanding physical properties. Not only are CNTs interesting materials of study for understanding fundamental physics in low dimensional systems [2] but also, they demonstrate material properties suitable for a vast range of applications [3]. Attachment of magnetic molecules on to functionalized CNTs is currently being widely explored for molecular spintronic applications [4]. The anisotropic spin of the metal ion confined on to carbon nanotube exhibits profound quantum properties such as quantum tunneling magnetization (QTM), spin blocking, etc. Although these materials are now widely studied, there is a serious need for the tailoring and tuning of CNT properties for the specific intended purpose. CNTs can be considered as macroscopic organic molecules and one of the main methods of augmenting specific functional groups is through chemical treatment [5]. However, the reproducibility of such methods remains a stumbling block in ensuring homogeneously prepared samples. This is mainly due to the lack of controlled degree of functionalization and as well the difficulty in analyzing the functionalization conditions. A spectroscopic technique that has seen great application into assessing the structural properties in carbon materials is Raman spectroscopy [6,7]. Although the combined effects of functionalization and disruption of the graphitic lattice can be difficult to interpret, recent work on elaborate deconvolution of the Raman spectra, emphasizing the relationship between first and second order harmonics, has led to interesting findings [8]. It was shown that the minor bands and satellite modes near the main G and D bands can collectively lead to a deeper understanding of the structural modification during chemical treatment. This work has also been used to address the shortfalls of the Koenig–Tuinstra relation when using the I_D_/I_G_ ratio in analyzing the structural deformity of multiwalled carbon nanotubes [9]. As shown in Figure 1a, these deconvolution schemes evaluate the weighting of minor modes adjacent to G and D bands (S + G_L_ and D_R_, D_L_ and G_R_).

Raman spectroscopy is sensitive to both long-range and short-range order and these side bands are argued to be related to two structural aspects of the nanotube surface modification: (i) the introduction of polycyclic aromatic hydrocarbon domains (PAH) [10,11] and (ii) polyacetylene and polyacrylic-like regions [12,13,14]. This interpretation has been used in a range of subsequent studies [15,16,17] and it is now clear that evaluating the playoff between PAH domains and the polyacetylene-like Raman signatures may be used as a more refined indicator of the degree of chemical functionalization than relying solely on the I_D_/I_G_ ratio [8].

Motivated by these interesting findings, we initiate the current work to better understand how the functionalization process modifies the nanotube surface and how such modifications influence the underlying physics, allowing us to tune CNT properties. In this work we extend previous studies [8,18] by investigating the degree of functionalization of MWCNTs in a systematic way. MWCNTs are functionalized using the conventional oxidative reflux reaction. To evaluate the effect of the reflux reaction on nanotube quality, a variation of reflux temperatures, as well reflux time dependence was used in the study and monitored through Raman spectroscopy. This allows us for tracking the effects of increased functionalization. In a final step, the functionalized MWNTs are loaded with a molecular magnet (Gd-DTPA) through a two-step synthetic chemistry route shown in Figure 1b. This route ensures that Gd-DTPA is preferentially attached to functional groups along the MWNT walls (Figure 1d), and thus the degree of functionalization would be directly related to magnetic properties of the complex system. The modified MWNTs are then assessed using Superconducting Quantum Interference Detector (SQUID) magnetometry, which allows for determining very small changes in the magnetic character across the studied samples. Through this combined study, we provide information on the fine tuning of magnetic properties through a refined and monitored chemical treatment. This information is expected to be significant in further developing CNT-based technologies such as novel spintronic devices [4], drug delivery [19] and nuclear magnetic resonance (NMR) contrast agents [20].

## 2. Results and Discussion

### 2.1. Raman Spectroscopy

Raman spectroscopy has been extensively used to determine the levels of structural disorder in sp^2^ carbon. This is generally analyzed in terms of the ratio of the I_D_/I_G_ phonon peaks originating from first order spectrum [21]. However, although this analysis is well established for graphite, graphene, double and single walled nanotubes, the Raman spectrum of the MWNTs is characteristically different and more closely resembles the spectra of chars and carbon blacks [9,22], thus an intense D band peak is always observed, even in unfunctionalized (pristine samples). As a result of this, the conventional I_D_/I_G_ ratio comparison is not as effective in giving an accurate quantitative idea of the level of disorder [9,22]. In fact, it has been shown that upon functionalization, the I_D_/I_G_ ratio does not necessarily change significantly [9] and thus it is important to investigate alternative mechanisms for determining variations in functionalization and disorder. Beyond these studies, it is known well accepted that upon increasing disorder and subsequently sp^3^ content, the interpretation of Raman spectrum can be categorized into various stages where the Kroenig–Tunistra interpretation breaks down [7].

Recently, a deconvolution scheme [8] has been developed that investigates the chemical functionalization degree (CFD) of the MWNTs by evaluating changes in the minor bands with respect to the main D and G bands: the area ratio of minor peaks with main D and G peaks, i.e.,
(1)$$\mathrm{CFD}=\Sigma \mathrm{area}\frac{\left(S,S_{R},D_{L},D_{R},G_{L},G_{V},D_{R}\right)}{\left(S,D\right)}\times 100$$

This is possible because as the degree of functionalization increases, the aromaticity of the sp^2^ lattice is gradually disrupted. Taking inspiration from this, we further probe into the evolution of the minor peaks and their correlation to aromatic structures. One of the main effects of the functionalization of the MWNTs is the broadening of the D and G bands which can be seen in their peak area variations. This is due to the formation of shoulder peaks that correspond to nanoscale Raman active sites along the tube. In this scheme, the D and G peaks were further deconvoluted to a cluster of between five and seven sub-peaks using Lorentz fitting [8,15,16,17]. The D band has two subpeaks on either side designated as D_L_ and D_R_ (D left and D right) at ~1250 cm^−1^ and ~1400 cm^−1^, respectively. At the left side of the spectrum a side band (S) at approximately 1200 cm^−1^ is also observed. In the G band region, two subpeaks can be located on either side of the main G peak, these are designated as G_L_ and G_R_ and located at ~1500 cm^−1^ and ~1610 cm^−1^, respectively. The occurrence of these subpeaks is rationalized to be due to two competing structural components, either polyacetylene/polyphenylene-type or poly aromatic hydrocarbon like (PAH) [10,11,12,13,14]. Due to this D_L_, D_R_ and G_L_ are used as confirmation of smaller size polyaromatic domains (PAH) whereas the S and G_L_ subpeaks are likely due to polyacetylene like structures. To ensure the best fitting procedure, we rely on two fitting criteria: firstly, the position of the bands in the 1100–1800 cm^−1^ are anchored by the corresponding bands in the second overtone region (2D) (shown in Figure 1). This is possible by ensuring that the fitting of the 2nd order bands correspond to the composite values of the first order modes, i.e., DS (D + S), D D_L_ (D + D_L_), 2D (D + D), D D_R_ (D + D_R_), D G_L_ (D + G_L_), DG (D + G) and 2G (G + G). Secondly only deconvolution analysis with R squared values lesser than 0.996 have been used. This ensures the most accurate fitting to the data and entirely consistent with previous reports [8].

The initial experiments are focused on the effect of oxidation time on the Raman modes. Previous reports have indicated that upon extended oxidation treatment, the most notable effects of the oxidation are an increase of the full width at half maximum (FWHM) of the G band [18], as well as a change in the effective percentage area of the sub-peaks [8]. As shown in Figure 2a–d, a clear change in the Raman spectra is observed when varying the oxidation time between 1 and 24 h. The most notable difference between samples oxidized for different durations is the enhancement of area of the satellite peaks. Figure 2a–d indicate how these satellite peaks initially start out with a relatively small percentage area when compared to the G and D bands. As the oxidation time increases these minor bands increase in area and overall weighting in the deconvolution process. As shown in Figure 2e, the I_D_/I_G_ ratio remains relatively constant across the oxidation times used in the study. This is contrasted with the ratio of the minor bands to main band ratio (indicated by red bars) which shows a much more pronounced change. This increase in relative Raman ratio emphasizes that using peak area instead of intensity gives a more sensitive metric in evaluating the functionalization. As shown in Figure 2f, along with the enhancement of the satellite minor bands an appreciable broadening of the FWHM of the G and D bands can be observed. It is widely accepted that the broadening of the FWHM of these bands is related to increase in disorder through physical disruption of the graphitic lattice and possibly of tube cutting [18]. As shown in Figure 2f, the G band shows a nearly linear increase with increasing oxidation times whereas the D band shows a more substantial initial increase. This is a strong indication of the different origin of these two peaks and how the oxidization affects them. The satellite peaks near the D band (D_L_ and D_R_) as well as the G_R_ minor band, are more closely related to the PAH-like domains [10,11] whereas the G_L_ and S bands are argued to indicate the presence of polyacetylene and polyphenylene like networks [12,13,14]. The enhanced broadening of the D band over the G band thus indicates that extended oxidization times are more strongly correlated with PAH domains across the surface of the functionalized nanotubes and not functional group networks. From the time dependence study, it is thus found that a minimum of 24 h is required to properly enhance the minor peaks. This is significant as it indicates that prolonged oxidization time, at least up to 24 h, does not necessarily lead to increased destruction of the tubes due to functional group attachment but rather from pitting effects and tube cutting associated with an abundance of PAH domains.

To further investigate the effects of reflux conditions on the functionalization of the nanotubes, a temperature dependence study is conducted. Here, the experimental parameter is the oxidization reaction temperature with fixed time (24 h). As shown in Figure 3a–d, once again a notable difference can be seen in the deconvoluted spectra. As shown in Figure 3e, the I_D_/I_G_ ratio shows a decrease with increasing reaction temperature whereas an evaluation of the ratio of minor and major bands (relative Raman ratio) shows a more pronounced and sensitive change, here decreasing abruptly after 60 °C. In contrast to the time dependent study, we note a significant shifting of the spectral weight between minor bands. This spectral shifting is most significant between the overlapping regions of the D and G bands (most likely due to the small energy difference in such modes). This spectral shift shows that higher wavenumber bands are enhanced (in both intensity and percentage area) as the reaction temperature is increased. This observation is significant as it indicates that an overall change in the type of functionalization may be occurring at higher temperatures. For the temperature dependent study, we identify a large increase in the weighting (area) of the G_L_ minor band (mirrored by an initial increase in D_R_ band) and eventual total broadening of the D_R_ band. As the G_L_ band is more closely related to polyacetylene and polyphenylene-like formation this observation indicates that elevated reaction temperatures may lead to an increase in out of plane functionalization group attachment and degradation of the PAH domains. This degradation is reflected in the broadening of the PAH related bands (most notable D_R_).

As indicated in Figure 3f, this variation in the effective area of the minor bands’ functionalization may be occurring at higher temperatures and allows for a new metric for evaluating the functionalization degree, here specifically referring to degradation of PAH domains in favor of polyacetylene-like networks (higher degree of functional group attachment). As shown in Figure 4a, using the ratio of (S + G_L_)/(D_L_ + D_R_ + G_R_) it is possible to evaluate the change in degree of functionalization in a much more refined manner than previously established and also to elucidate the difference in the effect of various chemical treatments used in the structural modification of the MWNTs (schematically shown in Figure 4b). Further, high resolution transmission electron microscopy (HR-TEM) was used to support these findings (Figure 4c,d). Long oxidation durations have promoted graphitic pits on the nanotube wall (seen from Figure 4c) while high temperature oxidation has promoted localized defects/nucleation sites (seen from Figure 4d) that would enhance the non-covalent attachment of metal complex on the nanotube. Here, we find that temperature dependent acid reflux reactions would yield more active oxygen functional moieties on MWNTs that in turn can be used in linking Gd-DTPA chelates.

Further evaluation into G band position has revealed that MWNTs treated at higher temperature showed G peak position shifting towards higher frequencies which can be explained that by forming higher percentage of oxygen functional groups and the respective electronegative effect on CNT structure influence the shift of G band (Appendix Aa,b). This is evident as the case of our samples treated at higher temperatures showed a higher percentage of Gd molecule attachment on the CNT surface and this further supports our claim where deeper analysis into the area of minor Raman bands can be used as new metric for evaluating degree of functionalization.

### 2.2. Modification of Magnetic Properties of Functionalized MWNTs Loaded with Gd-DTPA Chelate Attachment of Gd-DTPA

For a qualitative analysis to evaluate Gd-DTPA metal ligand complex attached on the carbon nanotubes, we have utilized time of flight–secondary ion mass spectroscopy (ToF-SIMS) surface characterization technique. Figure 5a, shows the increased Gd^+^ secondary ion counts with respect to increased reaction temperature (Gd^+^ secondary ion fragment is expected at 158 mass by charge (m/Q)). The 55 °C sample showed the lowest Gd^+^ ion counts (196128) while the 100 °C sample showed highest Gd^+^ ion counts (366552). This further supports our new adopted Raman analytic technique in evaluating chemical functionalization degree, where increased polyphenylene-like structures with respect to increased temperature reflux conditions would promote successful noncovalent attachment of the Gd^3+^ metal ion complex onto the nanotube surface. Although Boehm titration revealed more carboxylic groups present on the CNTs treated at 55 °C when compared with CNTs treated at 100 °C, we notice low Gd^+^ ion concentration in the 55 °C sample and high concentration in the 100 °C sample (Appendix Ab). This indicates that metal ion complex attachment on to CNT is more favored via carboxylic dimerization—oxalic functional groups (-C_2_H_2_O_4_) formed at higher reaction temperatures [23] and further polyene and polyphenylene-like aromatic structures formed on the nanotube surface treated at higher oxidation temperatures, act as active sites for successful metal ion attachment. Oxalic functional groups are often used as a good chelating agent in forming a stable complex with metal ions. The same has been observed in our samples, where Gd_2_O_3_ percentage used is high for the 55 °C sample but achieved a poor attachment of Gd^3+^ ion complex on the nanotube since most of the Gd^3+^ ions were removed during the sample washing process. In contrast for the 100 °C sample, we have seen most of the Gd^3+^ ion complex forming a stable noncovalent bond with the oxalic functional groups and remain intact on the nanotube surface even after multiple washing cycles.

Further, ToF-SIMS elemental mapping of Gd^+^ secondary ions (Figure 5b) reveal the even distribution of Gd^+^ metal ion on the CNTs. As seen from Figure 5b, the Gd^+^ ion brightness contrast is seen increasing from 55 °C to 100 °C treated CNTs. This suggest that temperature oxidative condition can be utilized in creating controlled chemical functionalization across the nanotube and respective magnetic metal ion complex loading percentages grafting on to the nanotube can be achieved. Thereby, a highly correlated spin system can be developed with desirable magnetic properties, where CNT would be acting as a bridging material to access the nanoscopic quantum properties.

### 2.3. SQUID Magnetometry

As deduced from the analysis of the Raman data, an increase in reaction temperature leads to disruption of the delocalized π-electron framework and an increase in polyene-like structures that dominate over PAH-like structures. This is significant as the magnetic properties of MWNTs are intimately linked to the conduction π-electrons of the graphitic lattice and thus any structural changes induced by the chemical treatment should also influence the magnetic properties of the MWNTs [24]. The diamagnetism of nanotubes has already been used to determine the interaction between inner tubes in MWNTS [25]. This diamagnetism is anisotropic in nature [26] and has been shown to be directly correlated to an Aharonov-Bohm effect [27]. Although predominantly diamagnetic, the effects of residual catalysts such as Fe, Ni or Co are known to lead to an enhanced paramagnetic signature in opposition to the diamagnetic framework [28]. In order to thus correctly analyze the magnetic signatures of the functionalized nanotubes, it is important to evaluate both high field as well as low field properties. At higher fields the paramagnetic contributions are saturated by the applied field and it is possible to establish the magnetic properties of the MWNTs. As shown in Figure 6a, there are indeed signatures of a pronounced difference in the magnetic moment with applied field for the samples prepared at different reaction temperatures as well as signatures of cohesive and remnant fields. The most notable observation is the change in the diamagnetic slope at higher fields. The negative slope here indicates the intrinsic diamagnetism of the MWNTs and as the reaction temperature is increased the slope increases until taking on a positive value for the most highly functionalized samples. This is shown in inset Figure 6a, where the derivative of the data is used to evaluate the change in slope across samples. The transition from a negative (diamagnetic) to a positive slope indicates the transition to a dominate paramagnetic response when highly functionalized.

As the reaction temperature increases, so does the structural degradation of the graphitic lattice. This leads to a decrease in the π-electron framework and subsequently the diamagnetic character and this corresponds to findings of the Raman analysis. These features are expected for a dilute or mixed magnetic material and again is linked to the interaction between delocalized electrons and the strong magnetic moment of the Gd-DTPA ligands, most likely due to a RKKY interaction [28]. As confirmed through the Raman analysis, the reflux at higher temperatures leads to the disruption of the graphitic lattice and thus to a more resistive material (localization of electrons near defect points or functional groups along surface).

This ensures that interaction via the conduction or itinerate electrons becomes less likely and therefore reduction of the coercive field. This has before been observed in magnetically decorated nanotubes and argued to be an indication of a transition to a superparamagnetic state when interaction between magnetic centers are completely prevented [29]. For our study, however, the system does not show the hallmark features of superparamagnetism [30] but more like a strongly spin-interacting system. As seen in Figure 6b, the temperature dependence of the magnetic moment for field cooled measurement (FC), no clear blocking temperature can be identified, ruling out the possibility of a purely superparamagnetic scenario. The susceptibility of Gd-DTPA and un-functionalized CNTs is provided as reference in the Appendix A. Furthermore, when plotting the inverse susceptibility as a function of the temperature, a pronounced linear region can be observed for much of the data, in accordance with the Curie–Weiss law:(2)$$\chi =\frac{C}{T-\theta }$$
where *C* is the Curie constant and θ is the Weiss temperature. It can be seen that the Curie–Weiss temperature takes on negative values (Figure 7b), a clear indication that antiferromagnetic coupling exists in the sample. Again, a trend can be observed between samples functionalized at different reaction temperatures. As expected, samples with a higher degree of functionalization correspond to lower Weiss temperatures. This observation alludes to the effects of the surface functionalization on the conduction electrons that mediate the magnetic interaction. As the reaction temperature increases so does the degree of functionalization. This ensures a suppression of the intrinsic diamagnetism of the nanotubes coupled to a higher Gd concentration ensuring an overall increase in paramagnetic character. As the magnetic character is not entirely dictated by the Gd-DTPA complex, but rather through the coupling between diamagnetic CNTs and Gd complex the data does deviate from the Curie–Weiss law at low temperatures. Such features are further indication of a highly correlated spin system and have before been identified as signatures of Kondo effect, where the point of deviation given as an indication of the Kondo temperature [30,31]. As shown in Figure 6b, we determine Kondo onset temperatures (T_K_) [31,32] of 132, 155, 188 and 202 K as a function on increasing Gd concentration. These findings are in agreement with the analysis of the Raman deconvolution study and shed light onto how structural degradation and magnetic loading can lead to enhanced spin correlations in functionalized carbon systems as recently demonstrated through the observation of the impurity Kondo effect [33,34,35]. Furthermore, these findings are significant as they help explain the role of itinerate electrons in mediating the interaction between spatially separated spin centers and thus may provide a mechanism for tuning the system from a superparamagnetic material to a highly correlated system with a definite spin texture. This is important as different applications of the nanotubes require specific magnetic character, for example NMR contrast agents rely on the superparamagnetic system however novel spintronic devices rely on highly correlated spin interactions [36,37,38].

### 2.4. CNT Magnetic Property Tuning

A combination of Inductively Coupled Plasma Optic Emission Spectroscopy (ICP-AES) and time-of-flight secondary ion mass spectrometry (ToF-SIMS) is used to confirm and quantify the successful attachment of the Gd containing chelate to the CNT surface. From ICP-AES, the weight percentage of the loaded Gd increases from 4.95% to 5.8% (Figure 7a) in correspondence to increased chemical functional degree. The Gd ion weight percentages concur with respective ^158^Gd counts measured from ToF-SIMS.

The combination of Raman spectroscopy and magnetic property analysis allows for a direct evaluation of the modified MWNT properties due to the chemical treatment and the magnetic impurity loading. Figure 7a,b shows that the secondary analysis done from all the measurements follow an increasing trend in the chemical functionalization degree, magnetic impurity loading percentage and Weiss temperature with respect to the increased acid reaction temperature. This increasing trend in Weiss temperature reveals a shift of the magnetic ordering from antiferromagnetic to ferromagnetic with respect to the increased Gd^3+^ ion loading which is made possible with increased MWNT chemical functionalization degree. We see a direct correlation between the CNT CFD and the Weiss temperature in the CNT(Gd-DTPA) hybrid complex. In the future theoretical analysis, we aim to quantity the relation between CFD and Wiess temperature.

## 3. Experimental 

### 3.1. Functionalization of MWCNTs

The conventional acid reflux technique [39] is used to add the organic functionalities to the surface of the MWCNTs. This involves a 1:3 volume ratio mixture (8–16 mL) of nitric acid to sulfuric acid as oxidizing agent to which 50–121 mg of pristine MWCNTs is added. The mixture is then refluxed under different conditions; Firstly, the reflux experiment is conducted at a temperature of 55 °C at a variation of reaction durations (1, 3, 12 and 24 h). Secondly, refluxing is conducted at a fixed duration of 24 h but at different temperatures (55, 60, 70 and 100 °C). Upon refluxing for 24 h the functionalized MWNTs are initially isolated from the viscous mixture by centrifugation followed by subsequent vacuum filtration and repeated rinsing with distilled water. The discarded distilled water is tested for pH neutrality to ensure no residual acid contamination or pH modulation. The filtered product is vacuum dried under nitrogen yielding the functionalized CNTs. Characterization of the functionalized material is completed before loading samples with the Gd-DTPA complex.

### 3.2. Gd-DTPA Chelate Synthesis

The Gd-DTPA chelate was prepared by taking Gadolinium oxide (Gd_2_O_3_) and Diethylene triamine penta-acetic acid (H_5_DTPA) in stoichiometric ratios (13.9 mmol Gd_2_O_3_; 27.6 mmol H_5_DTPA) dissolved in 60 mL of distilled water. The turbid solution was magnetically stirred for 20 h at a temperature of 90 °C. Complete dissolution after reaction time of 3 h with continued stirring was evidenced by the disappearance of turbidity form to a clear, transparent solution indicative of Gd-DTPA complex formation. Upon cooling to room temperature, the solution is filtered using a 0.22 μm PVDF membrane filter. The filtered solution is allowed to evaporate under ambient conditions yielding white crystals of the Gd-DTPA complex upon slow evaporation. The crystalline product is collected by filtration and further evaporation resulted in a white crystalline precipitate that was collected and dried under reduced pressure for 4 h duration.

### 3.3. CNT-Gd-DTPA Complex Synthesis

MWCNTs with pre-determined functional (-COOH) group percentage and Gd-DTPA chelate were weighed stoichiometrically in ratios of 1:1 and mixed in 100 mL of distilled water. The mixture solution was subjected to magnetic stirring for a period of 24 h at room temperature. The resultant mixture solution was filtered using a 0.22 μm PVDF membrane filter and vacuum dried under reduced pressure. The collected dried powder was taken for further analysis.

### 3.4. Characterization

HR-TEM was used to image and analyze the structural changes along the nanotube. Magnetic molecules loaded MWNT were visualized with HR-TEM and find that an even distribution was achieved across the nanotube. Formation of carboxylic groups along the nanotube were identified using FT-IR and Boehm titration.

Raman spectra were acquired using the 514.5 nm line of a Lexel Model 95-SHG argon ion laser as excitation source and a Horiba LabRAM HR Raman spectrograph equipped with an Olympus BX41 microscope attachment. The incident light was focused onto the sample using a 100× objective (N.A. = 0.90) and the backscattered light was dispersed via a 600 lines/mm grating onto a liquid nitrogen cooled CCD detector. Data were acquired using LabSpec v5 software. The power at the sample was ~0.4 mW. Raman spectra for each sample were acquired for integration times of 60 s.

Quantification of Gd content in the hybrid complex was analyzed using ICP-AES (ICAP Thermofischer Scientific) and ToF-SIMS. Gd-DTPA loaded CNTs were analyzed by ToF-SIMS, using a LYRA3 focused ion beam scanning electron microscope (FIB-SEM) system (TESCAN, Brno, Czech Republic) with a C-ToF module provided by TOFWERK (Thun, Switzerland). The instrument uses Ga^+^ primary ion beam and is operated at 30 keV with ion beam current of 1nA and analysis is done in positive ion mode. The Ga^+^ ion beam was rastered over an area of 25 × 25 μm^2^ and the mass spectra data were collected for 4 min for all the samples. Cts/ToF-extraction in mass spectrum is average signal (count) per every pixel. This means one count represents one detected Gd secondary ion. Hence, the counts were evaluated as area under peak × number of all pixels.

Magnetic measurements were performed using a MPMS-Quantum Design SQUID magnetometer operating between 1.8 and 300 K at field (*H* = 100 Oe). Susceptibility measurements were performed on thoroughly washed Gd-DTPA (80 mg).

## 4. Conclusions

We have demonstrated that the route of oxidative functionalization of MWNTs under mild conditions can lead to a more refined and controllable method for tailoring the physical properties of the MWNTs. We also demonstrate that recent progress made in the interpretation of the deconvoluted Raman spectrum of MWNTs [7] can be vital for determining the extent to which the functional groups are attached to the MWNTs. Using the conventional seven peak deconvolution scheme of the G and D band regions, we establish that the ratio of minor bands (S + G_L_)/(D_L_ + D_R_ + G_R_) which are related to two distinct types of Raman active structural components (chain-like polyacetylene/phenylene-type or PAH-type clusters) can be a new and useful metric for establishing the structural modification under oxidative functionalization. This is significant as previous studies have shown that although the famous Koenig–Tuinstra relation is appropriate for other nanostructured carbon systems, it does not properly explain the Raman spectra of MWNTs. Furthermore, we establish that functionalization of the tube surface and subsequent attachment of magnetic Gd-DTPA consequently the changes in magnetic character of the system. This is because the magnetic properties are intimately linked to the Gd-DTPA concentration and quality of nanotube lattice (and thus the degree of functionalization). As the utilization of MWNTs for various applications necessitates the functionalization of the outer walls, it is imperative to ensure a controlled way of functionalization as well as systematic analysis of such modifications. Here, we report that coupling of SQUID magnetometry and Raman spectroscopy analysis would allow for a sensitive and systematic method in engineering a highly correlated spin-interactive system with MWNT as a host material that has potential applications in spintronics and quantum information processing.

## Figures and Tables

**Figure 1 molecules-26-00563-f001:**
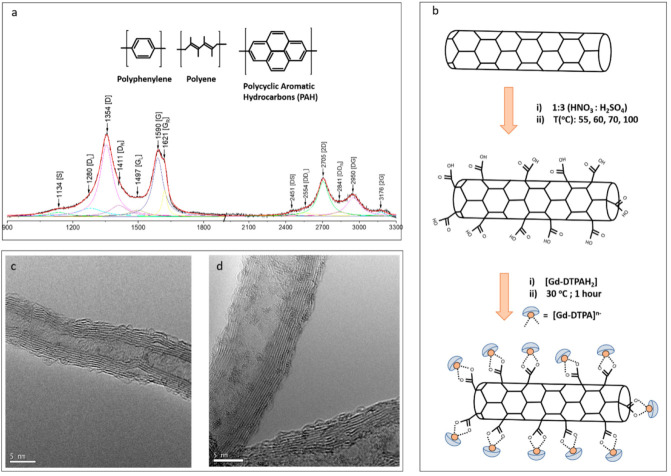
Raman spectroscopy is used to evaluate the effect of the mild oxidative functionalization technique. The D and G bands are deconvoluted to seven distinct Lorentz curves whose positions are anchored by the relevant second harmonic modes in the 2D region: (**a**) Following convention, we further distinguish between the minor modes based on their respective origin, either polyaromatic hydrocarbon type (PAH) or out of plane vibrations similar to polyene and polyacrylic structures. This allows for the tracking of the evolution of functionalization as a function of experimental parameters such as the oxidation time as well as reaction temperature. (**b**) Scheme for attaching a molecular magnet (Gd-DTPA) on to MWNTs through functional groups. (**c**,**d**) High resolution transmission electron microscopy (HR-TEM) image of pristine MWNT and CNT (Gd-DTPA).

**Figure 2 molecules-26-00563-f002:**
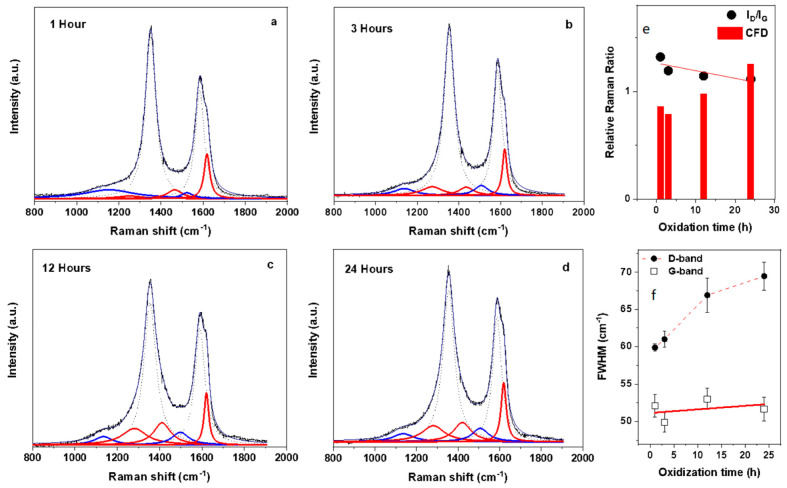
Raman spectroscopy used to evaluate the effect of extended time of oxidation. (**a**–**d**) The effect of oxidation time used to functionalize the MWNTs has a pronounced modification of the Raman spectrum, an increase in oxidation time results in the amplification of the satellite peaks relative to the main D and G modes. (**e**) The I_D_/I_G_ ratio is conventionally used to evaluate levels of disorder in graphitic type materials but does not capture the full effect of increasing functionalization. The ratio here shows a linear decrease with increasing oxidation time whereas evaluation of the effective area of the deconvoluted peaks is more sensitive to the variation in oxidation times and here shows a more pronounced rate of change than the I_D_/I_G_ ratio. (**f**) The increase in the effective area of the satellite peaks is accompanied by an increase in the FWHM of both the G and D bands.

**Figure 3 molecules-26-00563-f003:**
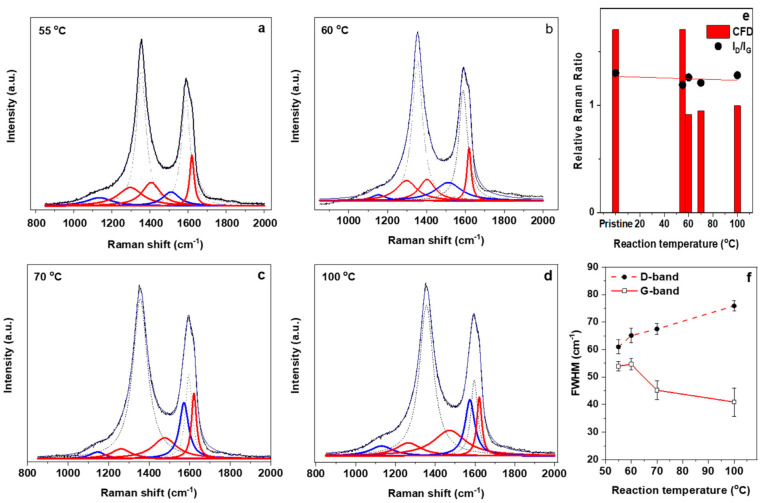
Raman spectroscopy is further used to evaluate the effect of temperature on the mild oxidative reflux functionalization technique. As shown in figure (**a**–**d**), the deconvolution method shows that a distinct spectral shift occurs, particularly to the shoulder bands nearest to the D and G overlap. As the temperature increases the D_R_ and G_L_ peaks show the largest increase. This shifting of dominance with respect to relative area can be used as a metric to track the increase in out of plane vibrational modes related to the functional group attachments. (**e**) As before, the change in effective area of the satellite peaks is more sensitive to the structural changes than the I_D_/I_G_ ratio. (**f**) The main reason for the change in the chemical functionalization degree (CFD) is due to a change in weighting of the D_R_ and G_L_ modes.

**Figure 4 molecules-26-00563-f004:**
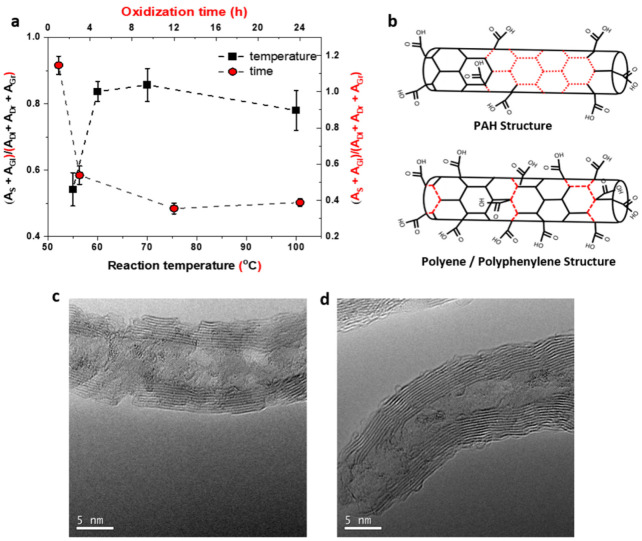
Evaluation of the ratio of minor bands grouped by predicted structural origin (either polyphenylene or PAH). (**a**) The stark variation between oxidation time and reaction temperatures shows that these two different experiments favor the formation of different functionality. The relative Raman ratio of increasing oxidization time shows a sharp decrease, then saturates, this is related to decrease in area of the S and G_L_ bands relative to the D_L_, D_R_ and G_R_ bands. When varying the reaction temperature, increased oxidation temperatures lead to an increase in the relative Raman ratio (increase in S and G_L_ over the D_L_, D_R_ and G_R_ bands). (**b**) This information can be related to the quality of functionalization of the tubes as the S and G_L_ bands are more closely related to polyphenylene like structures whereas the D_L_, D_R_ and G_R_ are related to PAH clusters (schematically shown on the right). The Raman spectroscopy findings are supported by high resolution transmission electron microscopy (HR-TEM) studies, as shown in (**c**), the longer oxidation times do lead to pitting effects and formation of graphitic segments that give rise to the PAH Raman signatures, compared to the temperature dependent study. (**d**) Where polyphenylene-like signatures are dominant in the temperature dependent oxidized nanotube samples.

**Figure 5 molecules-26-00563-f005:**
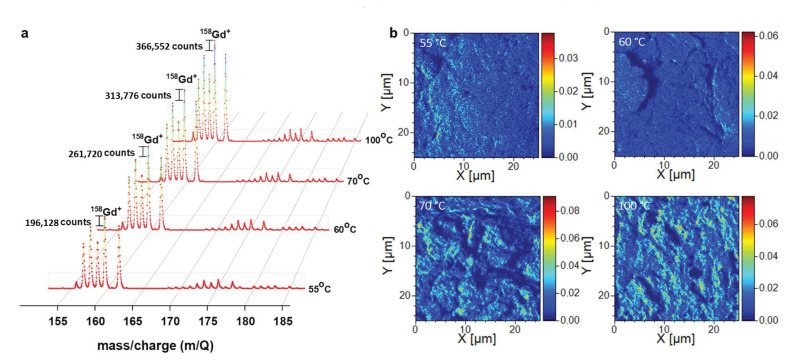
(**a**) ToF-SIMS spectra shows increased Gd counts for samples with a higher degree of functionalization (due to the increased reaction temperature). (**b**) ToF-SIMS element distribution maps reveals the evenly distribution of Gd complex on the carbon nanotube (CNT) surface and further increment of Gd percentage in the CNTs with respect to increased reaction temperature can also be visualized (bright contrast increment).

**Figure 6 molecules-26-00563-f006:**
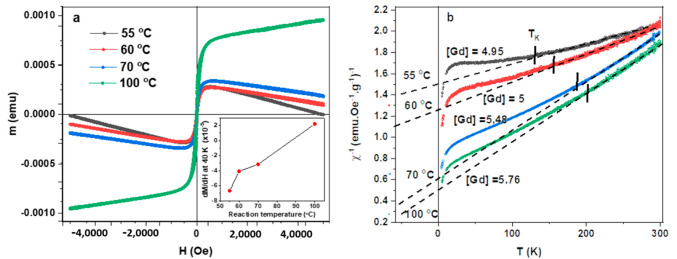
(**a**) The intrinsic diamagnetic response of the nanotubes is reduced with increasing temperature; indicating disruption of the conduction π-electron network as more Gd-DTPA is attached to the nanotubes, evident through the increase in slope of high field data. Inset—The slope increases with increasing reaction temperature at higher fields. (**b**) Temperature dependence of the magnetic susceptibility indicates an interactive magnetic system with dominant antiferromagnetic interaction. The Weiss temperature is extracted by fitting the Curie–Weiss law and shows a pronounced increase in value with increasing reaction temperature. The breakdown of the linear behavior at low temperatures signifies the modification in spin-interactions due to the functionalization process and can be linked to the onset of a Kondo effect with Kondo temperature denoted by T_K_.

**Figure 7 molecules-26-00563-f007:**
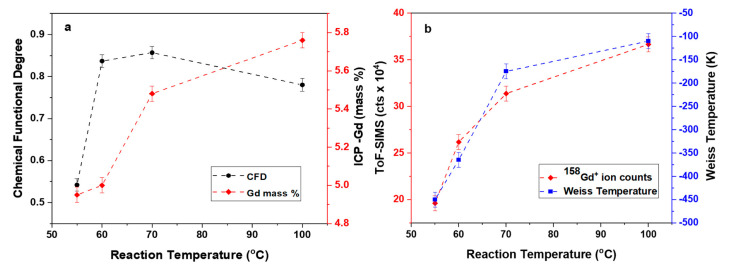
Tuning magnetic properties of MWNT coupled with Weiss temperature by changing functionalization degree and respective magnetic molecule loading percentage. Change in the oxidation reaction temperature of CNTs has a direct influence on their (**a**) chemical functionalization degree, (**b**) magnetic molecule (Gd-DTPA) loading percentage (ICP-AES and ToF-SIMS count) and Wiess temperature.

## Data Availability

The data presented in this work will be made available on request from the corresponding author.

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
