# Peer review of "Tuning Magnetic Properties of a Carbon Nanotube-Lanthanide Hybrid Molecular Complex through Controlled Functionalization"

_molecules, 2021, doi:10.3390/molecules26030563_

Round 1

Reviewer 1 Report

The paper reports on the functionalization of MWNT with Gd to prepare a material suitable for spintronics. The paper is well structures and first demonstrates the loading of Gd as a function of the functionalization treatment and subsequent magnetic characterization.

The first part on the paper focusses heavily on a recent evaluation procedure of the Raman spectroscopy which replaces the Kroenig-Tuinstra relation. I have the following suggestions and questions:

  1. It would simplify the reading of the paper if the peaks used for the calculation of the CFD would be labelled in Figure 1 or Figure 2.
  2. The FWHM of the D-band seems to scale better with the later Gd content (measured by SIMS), than the CFD and Id/Ig ratio. Can the authors please comment on this aspect? One would expect that this is accompanied by a shift of the G-Band. Can you please include the position of the G-band in the manuscript?
  3. The modification represents a surface treatment and the “pitting effect” effects approximately ½ the shells of the MWNT. Do the authors assume a homogeneous functionalization of all MWNT shells?
  4. Figure 4c;d – I recommend to include a TEM image of only the temperature defect (polyphenylene structure) prior to functionalization.
  5. The Id/Ig ratio relatively large and decreases with oxidation time. This indicates some contribution of sp3 coordination which is outside the main assumption of the Kroenig-Tuinstra relation. Please refer to Ferrari paper (doi: 10.1016/S0925-9635(01)00730-0 ) What assumptions does your evaluation procedure rely on?
  6. Please add the Raman integration time in the experimental section.

Reviewer 2 Report

The presented manuscript reports on the characterization of a hybrid system composed of carbon nanotubes and Gd-based complex. The paper is written in a chaotic way and English needs some improvement.
The most important findings are related to the tuneable magnetic properties, however, the data are not processed correctly. First of all, the Gd concentration should be determined and the magnetic data should be normalized by Gd content. Moreover, the magnetic properties of pure Gd-DTPA, as well as carbon nanotubes, should be measured and presented as a reference.
The Weiss temperature was determined based on FC susceptibility measurements, but as it is shown in Fig. 6b the plots are not linear- the diamagnetic contribution was not subtracted. Moreover, the authors don't inform about the fitting range. There is a lack of the spin state of Gd ions and their interactions in the discussion of magnetic properties.

Reviewer 3 Report

The paper focuses on molecular magnets attached to carbon nanotubes (CNT). This work demonstrates the chemical functionalization degree associated with molecular magnet loading can be utilized for controlled tuning the magnetic properties of a CNT-lanthanide hybrid complex. CNT functionalization degree was evaluated by interpreting minor Raman phonon modes in relation to the controlled reaction conditions. Inductively coupled plasma mass spectrometry, time–of–flight secondary ion mass spectrometry and SQUID measurements were used to elucidate the variation of magnetic character across the samples. This controlled Gd-DTPA loading on CNT surface has led to a significant change in the nanotube intrinsic diamagnetism, showing antiferromagnetic coupling with increase in the Weiss temperature with respect to increased loading. The results are very interesting and new. The paper is well written. Therefore, I would like to recommend publication after minor revisions. Figure 6a should be improved: the indications of temperatures should be shifted up to the second quadrant of this figure, while the coordinates of the inset should be more clearly shown.

Round 2

Reviewer 2 Report

The revised version of the manuscript is improved in comparison to the previous one. The authors have answered all of my questions and doubts. It can be accepted for publication in the present form.